# Large thermoelectric power factor from crystal symmetry-protected non-bonding orbital in half-Heuslers

Jiawei Zhou [1], Hangtian Zhu[2], Te-Huan Liu[1], Qichen Song [1], Ran He[3], Jun Mao [2,4], Zihang Liu[2], Wuyang Ren[2], Bolin Liao[5], David J. Singh[6], Zhifeng Ren[2] & Gang Chen [1]

Modern society relies on high charge mobility for efficient energy production and fast information technologies. The power factor of a material—the combination of electrical conductivity and Seebeck coefficient—measures its ability to extract electrical power from temperature differences. Recent advancements in thermoelectric materials have achieved enhanced Seebeck coefficient by manipulating the electronic band structure. However, this approach generally applies at relatively low conductivities, preventing the realization of exceptionally high-power factors. In contrast, half-Heusler semiconductors have been shown to break through that barrier in a way that could not be explained. Here, we show that symmetry-protected orbital interactions can steer electron–acoustic phonon interactions towards high mobility. This high-mobility regime enables large power factors in half-Heuslers, well above the maximum measured values. We anticipate that our understanding will spark new routes to search for better thermoelectric materials, and to discover high electron mobility semiconductors for electronic and photonic applications.

[1] Department of Mechanical Engineering, Massachusetts Institute of Technology, Cambridge, MA 02139, USA. [2] Department of Physics and Texas Center for Superconductivity, University of Houston, Houston, TX 77204, USA. [3] IFW Dresden, Institut für Metallische Werkstoffe, Helmholtzstraße 20, 01069 Dresden, Germany. [4] Department of Mechanical Engineering, University of Houston, Houston, TX 77204, USA. [5] Department of Mechanical Engineering, University of California, Santa Barbara, Santa Barbara, CA 93106, USA. [6] Department of Physics and Astronomy, University of Missouri, Columbia, MO 65211, USA. Correspondence and requests for materials should be addressed to Z.R. (email: zren@uh.edu) or to G.C. (email: gchen2@mit.edu)

In the last decades there have been growing efforts to access high electron mobility materials, which are essential for diverse applications ranging from solar cells to transistors. Conventional wisdom targets small effective masses (e.g., InSb[1]) to achieve high mobility. The recent discovery of graphene[2] and its three-dimensional analogs, topological Dirac semimetals[3–5], established that record-high mobility can also emerge from topologically protected Dirac bands with linear energy-momentum relations. This high mobility is a consequence of large electron velocities, a signature that reflects the electronic band structure. Another often-neglected degree of freedom that governs the intrinsic limit of electron mobility above cryogenic temperatures is the exploration of scattering probabilities—specifically, the electron–phonon interaction (EPI)[6]. Despite having fundamental understanding of how EPI regulates superconductivity[7] and polaron formation in conductive polymers[8], tuning its strength for high charge mobility is still uncharted territory.

Recently, electron transport mechanisms have also attracted growing interest in the context of thermoelectric materials[9]. Thermoelectric devices directly convert heat into electricity[10]. Their efficiency, as characterized by the material's figure of merit $zT = \frac{\sigma S^2 T}{\kappa}$, is largely limited due to the competing nature of the transport properties involved[9]: $\sigma$, the electrical conductivity, $S$, the Seebeck coefficient, and $k$, the thermal conductivity. Significant progress in reaching higher values of zT has been achieved by greatly reducing $k$[11–13]. Improving the power factor ($\sigma S^2$), however, has been a much more challenging task. One hitherto widely-adopted strategy is the "band engineering" approach[14–16], wherein the electronic band is tuned by composition or temperature to facilitate a large (or sharply changing) density-of-states, which favors enhanced Seebeck coefficients. The Seebeck coefficients can also be raised through exploitation of complex band structures[17] or by crossover between carrier transport regimes[18]. Despite the past progress, reaching large power factors has remained demanding because a high electrical conductivity must be accompanied by a large Seebeck coefficient, yet these two properties are often in competition (e.g., a small effective mass favors high $\sigma$, but opposes a large $S$). On the other hand, the EPI is a sort of control knob for electron mobility, and tuning it has only a weak influence on the Seebeck coefficient. The EPI is, therefore, the ideal choice for resolving the conundrum of high conductivity and large Seebeck coefficient. Even so, identifying materials with specific EPI has proven to be a difficult task due to lack of understanding of the connection between EPI and crystal structures.

In this work, we uncover the intimate link between EPI and orbital interactions facilitated via crystal symmetry, by investigating electron transport in half-Heusler systems in unprecedented detail. Half-Heusler materials—systems bearing a cubic crystal structure with three atoms per unit cell[19]—are well-known for their high-power factors (including the highest value ever reported in bulk semiconductor systems above room temperature[20]), but the origin of these power factors has been unclear so far. We study fifteen stable half-Heusler compounds based on past work[21]. By using first principles computational tools (Methods) we reveal that the high-power factors of half-Heusler compounds originate from a strong suppression of electron–acoustic phonon couplings, which contradicts the common belief that the electron transport in such materials is limited by acoustic phonon scattering[19]. Through orbital analysis we determine that this weak acoustic phonon scattering is enabled by the non-bonding orbitals—electron states mostly resembling single atomic orbitals with vanishing bonding (anti-bonding) character—at the band edge. While past simulations mostly rely on a constant relaxation time approximation[22], we employ the Wannier interpolation scheme for EPI[23], which enables the study of electron transport mechanisms without ad hoc parameters (Methods). A key insight we provide is that the vanishing bonding (or anti-bonding) character can be protected by the crystal symmetry.

## Results

**Electron–phonon interaction.** EPIs are often written in the form of coupling matrix: $\langle \psi_{\mathbf{k}} | \partial_{\mathbf{q}} V | \psi_{\mathbf{k+q}} \rangle$, representing the transition probability between electron states mediated by one phonon[23]. Figure 1a, b shows calculated EPI matrices associated with (longitudinal) acoustic and optical phonons for exemplary materials in their conduction bands. For small phonon wave vectors, the EPI's are well approximated by straight lines, with the slope being defined as the deformation potential (see Methods for details), which characterizes the electron–phonon coupling strength. The polar interactions are excluded for now because thermoelectric materials are often so heavily doped that any polar effect will be strongly screened by the free carriers (Supplementary Note 2 and Supplementary Fig. 15).

Deformation potential and effective mass are the two key parameters that affect the electron mobility. In practice, given the effective mass, one can extract the deformation potential from experimentally measured mobility, if other scattering mechanisms can be excluded[24]. For half-Heusler materials, a low effective deformation potential has been obtained based on experiments[19,25]. As polar optical phonon scattering is strongly screened at high carrier concentrations, past work has attributed the intrinsic origin of such a low deformation potential value to acoustic phonons[19,20,25], assuming acoustic-phonon-limited mobility. However, as seen in Fig. 1c, d, acoustic deformation potentials (~1 eV) for conduction band edge states in half-Heuslers are much smaller than the optical ones (~4 eV), suggesting that the electron–acoustic phonon coupling in half-Heuslers is in fact very weak—much weaker than had been assumed. We have also computed the deformation potentials for valence band edge states as shown in Supplementary Fig. 1. The results again indicate that acoustic phonon deformation potentials in general are much lower than optical phonon ones. Accordingly, the experimentally extracted deformation potential also involves optical phonons, in contrast to the traditional view that it is the acoustic phonons that limit the electron transport in half-Heuslers.

**Crystal orbital analysis.** The surprisingly weak electron–acoustic phonon coupling stimulates us to query the reason behind. As originally discussed by Bardeen and Shockley[26], the acoustic deformation potential, derived from electron–phonon couplings, can be linked to the electron energy level change as the lattice is strained. In light of this picture, one can explore the electron–acoustic phonon couplings from a standpoint of orbital interactions that create such energy levels. At equilibrium lattice spacing, atomic orbitals from different atomic sites interact and form bonding and anti-bonding states. As we expand the lattice, due to the weakened coupling between atoms, the bonding state will increase its energy while the anti-bonding state lower the energy, which translates to positive deformation potential for the former and negative values for the latter. A significant bonding (or anti-bonding) characteristic at the band edge normally implies a large acoustic deformation potential, as often found in many good thermoelectric materials[20].

Half-Heuslers (denoted ABC, like in ZrNiSn), however, reveal a pronounced distinction. Here, we employ the point group symmetry (at the band edge) to categorize the orbitals according to their group representations (Fig. 2e, Methods) based on a tight-

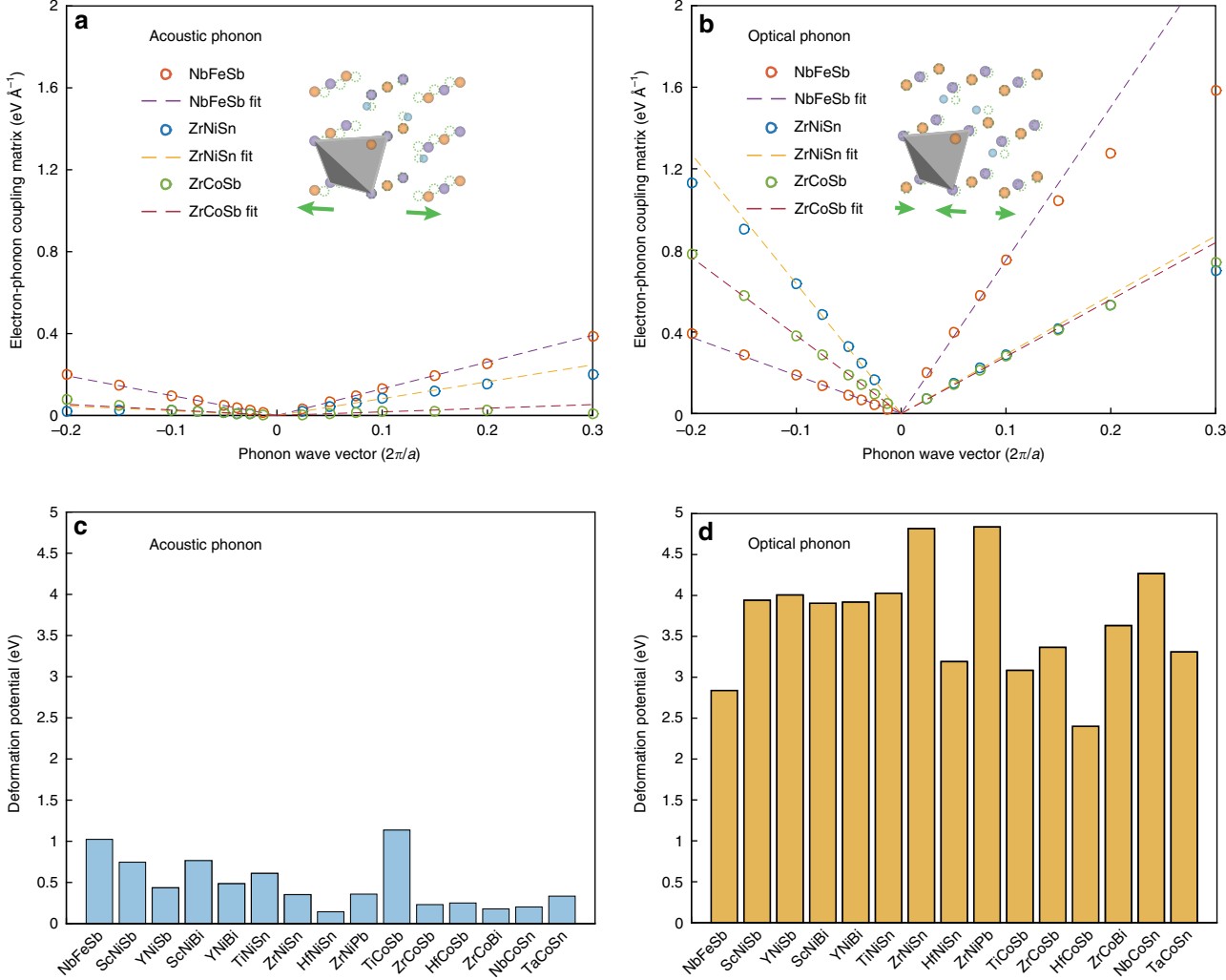

**Fig. 1** Electron–phonon interaction and deformation potential. **a**, **b** Electron–phonon coupling matrix along high-symmetry lines. The coupling matrices are calculated along two different directions in the Brillouin zone for **a** longitudinal acoustic phonons and **b** optical phonons, respectively, with the initial electron state located at the conduction band edge ($\mathbf{k}_X = (0, 0, 2\pi/a)$, where $\alpha$ is the lattice constant) and the final electron in the same band (intra-band coupling). The insets in **a** and **b** illustrate the lattice distortions corresponding to acoustic and optical phonons. **c**, **d** Averaged deformation potential for acoustic and optical phonons. The deformation potentials are extracted as the slopes from **a**, **b**, for the two different directions (one is parallel with $\mathbf{k}_X$ while the other is perpendicular), defined as $\Xi_\parallel$ and $\Xi_\perp$. The deformation potential generally depends on the directional angle of the phonon state, but for simplicity we calculate averaged deformation potential defined as $\overline{\Xi} = \Xi_\parallel^{1/3} \Xi_\perp^{2/3}$

binding formalism. In a tight-binding model (equivalently known as linear combination of atomic orbitals, or LCAO), one first defines crystal orbital as a superposition of atomic orbital states (Methods, Eq. 3). The electron energy levels are then solved by turning on the interactions between such crystal orbitals. Owing to the different orbital types and spatial arrangement of atoms, the orbital interaction energies can vary from positive to negative, making an intuitive visualization of the final energy levels difficult. The power of symmetry analysis based on the group theory is its ability to categorize these crystal orbitals into different classes based on their transformations under symmetry group operations, which are known as symmetry representations. Orbitals belonging to different representations do not interact with each other, thereby greatly simplifying the analysis. We emphasize that such symmetry analysis does not depend on the details of the orbital interactions, and therefore is also not limited to any particular orbital type. The energy levels resulting from interactions considering different groups of representations can be illustrated using an orbital diagram, as shown in Fig. 2e–f.

Remarkably, for half-Heuslers, one finds the conduction band edge state (X-point) is characterized by a distinct non-bonding representation $B_1$ (e.g., in NbFeSb, as shown in Fig. 2a), corresponding to $d_{x2-y2}$ crystal orbital at site A (Fig. 2e). Based on the picture we have discussed, such crystal symmetry-protected non-bonding orbitals should in principle have zero deformation potential. Another example of non-bonding orbital is given by the valence band edge state (L-point) in NbFeSb (Fig. 2f). While this state (representation E) no longer derives from a single atomic orbital, its projected density-of-states indicates that it consists of majorly $d$ orbitals from site B (Fig. 2b). Based on this, we call this a nearly non-bonding orbital, because if it had interacted with nearby atomic orbitals with lower (higher) energies (for example $p$ orbitals from site C), it would have acquired an appreciable amount of projection on that orbital with significant anti-bonding (bonding) character. The predominant projection on B's $d$ orbitals implies that this state closely resembles a non-bonding state, and, therefore, is expected to also have small deformation potential. In this case, the nearly non-bonding characteristic is partly protected by the crystal

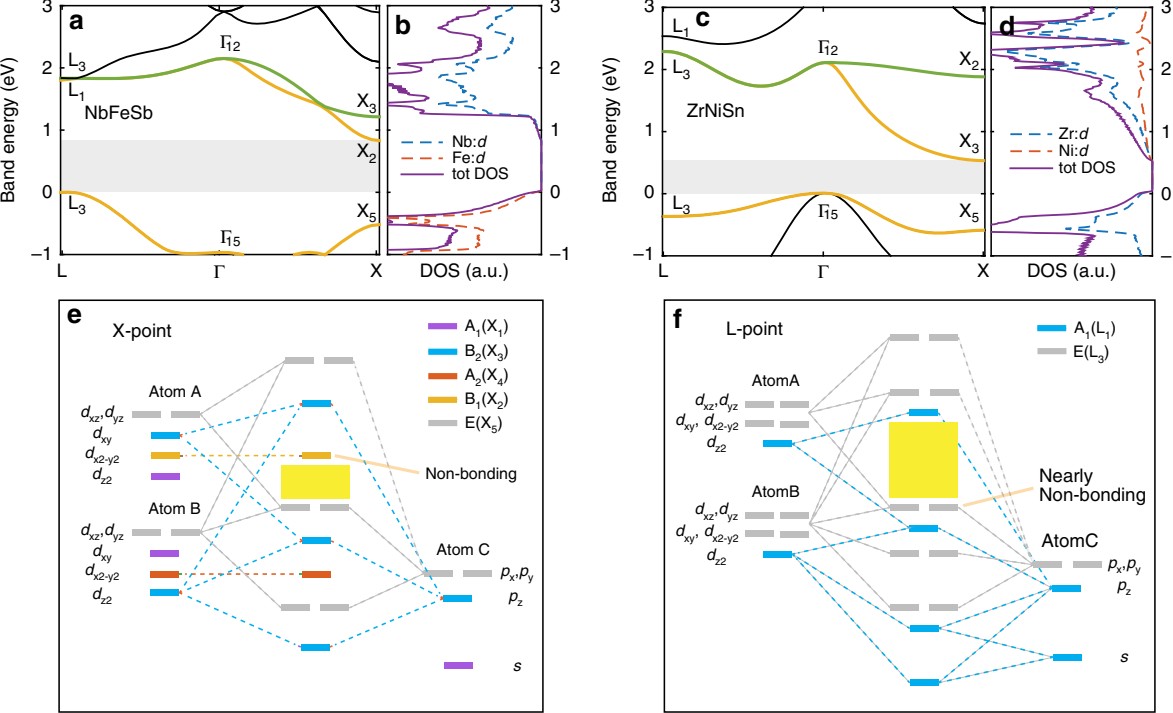

**Fig. 2** Crystal orbital analysis. **a–d** Band structures and corresponding projected density-of-states near conduction and valence band edge for NbFeSb and ZrNiSn. The band states at high-symmetry point (X, Γ, and L) are labeled by their corresponding group representations, obtained based on the first principles wavefunctions. Shaded areas in part **a**, **c** denote the band gap. **e**, **f** Simplified crystal orbital diagram for a prototypical half-Heusler compound, at X-point and L-point, corresponding to point group symmetry $D_{2d}$ and $C_{3v}$, respectively. The composition is denoted by ABC (B occupies the tetrahedral site, with A, B usually being the transition metals). To avoid complication, we have only employed a minimal basis set where $d$ orbitals at sites A, B, and $s$, $p$ orbitals at site C are included. Orbitals belonging to different irreducible representations are denoted by different colors. Different $p$ ($d$) orbitals are written apart just to avoid the overlap, while in practice they have the same energy. The yellow regions in part **e**, **f** denote the band gap

symmetry—orbitals with representations $A_1$ do not interact with the valence band edge state of NbFeSb.

Similar analysis holds for other half-Heusler compositions. For example, in ZrNiSn, the valence band edge state (at Γ point) characterized by $T_2$ representation represents a nearly non-bonding orbital (Supplementary Fig. 2), which is also confirmed by its predominant projection on Zr's $d$ orbitals (Fig. 2d). On the other hand, ZrNiSn's conduction band edge state corresponds to $B_2$, while in our model it has higher energy than the band edge state $B_1$ (Fig. 2b). This suggests that the anti-bonding $B_2$ state has been pushed downwards in energy, probably due to interactions with orbitals at even higher energies that are not included in our simplified diagram. Without such interaction, the anti-bonding $B_2$ level would have had large deformation potential. Its interaction with a higher energy state mostly introduces a bonding contribution to the $B_2$ state, which cancels part of its originally dominant anti-bonding character, rendering the final $B_2$ state at the band edge to be nearly non-bonding (or weakly bonded). In Supplementary Note 3 (also see Supplementary Figs. 4 and 5), by adding an additional orbital that interacts with the highest energy level within the $B_2$ representation based on a three-level system, we show that the energy lowering of $B_2$ level indeed leads to a cancellation of bonding and anti-bonding interactions, together with a much smaller deformation potential than what it would have if $B_2$ was a predominantly anti-bonding level. Again, we note that such non-bonding level is possible because the symmetry protection has prevented its interaction with many other states of different representations, especially those at similar energies. Had they interacted with each other, we would have most probably a band edge state that carries

significant bonding or anti-bonding character, and thus large deformation potential.

In line with the molecular orbital theory[27], we thus summarize the non-bonding orbital in a solid as one that has dominant contribution from a single orbital, when its interactions with other orbitals are mostly symmetry forbidden. Different from molecular orbital theory, the orbital states for solids are delocalized as they are superpositions of atomic orbitals throughout the lattice, which are called crystal orbitals[28]. Here, we want to clarify that the symmetry-protected non-bonding orbitals are not limited to $d$ orbitals as we have seen in half-Heuslers, as the symmetry analysis does not impose any constraint on the orbital type. The spatial arrangement of atoms and the orbitals together determines the group representation of a given crystal orbital. In fact, our symmetry analysis in $Mg_2Sn$ has shown that its conduction band edge state (at X-point), known to be predominantly Mg 3s state[29], represents another example of symmetry-protected non-bonding orbital. If we consider only $s$ and $p$ orbitals, the conduction band edge turns out to be the only state that is characterized by $B_{2g}$ representation (within $D_{4h}$ point group). Generally, as long as the symmetry allows, $s$ or $p$ orbitals can also create non-bonding states. However, due to our current numerical limitations (our density functional theory (DFT) calculation predicts $Mg_2Sn$ to be a metal instead of a semiconductor), we do not cover its transport properties in this work. In passing, we also mention that one can quantify the non-bonding character through the crystal orbital Hamilton population (COHP) analysis, which measures the degree of bonding/anti-bonding in an energetic scale[30] (Methods). Supplementary Fig. 3 shows that the small acoustic deformation potentials seen in half-Heuslers indeed match the small COHP values, which

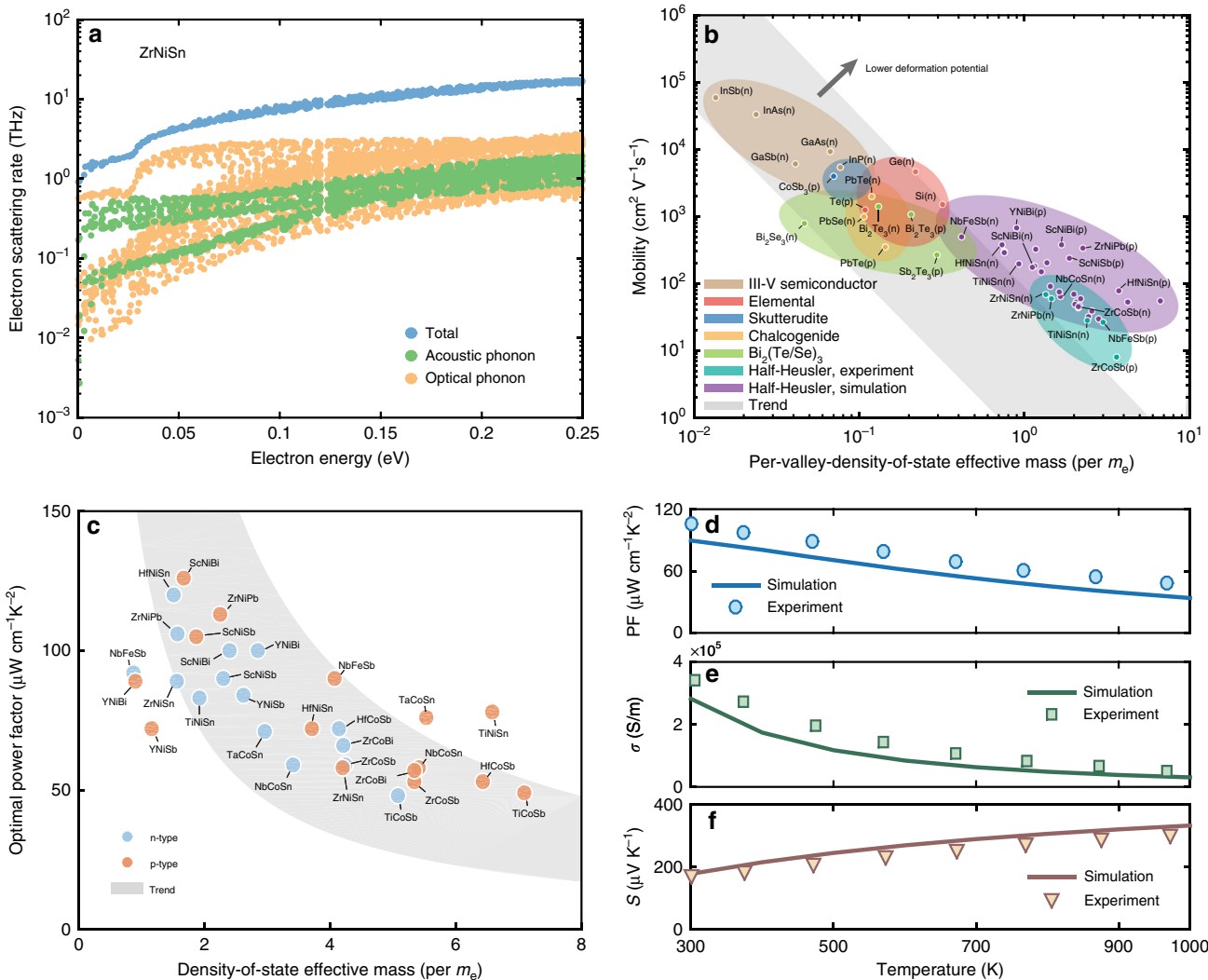

**Fig. 3** Electron dynamics and transport. **a**, **b** Electron scattering rates decomposed into contributions from different branches for ZrNiSn at room temperature with a carrier concentration of $10^{18}$ cm$^{-3}$. **b** Predicted intrinsic mobility for half-Heuslers compared with experimental values from different material families (including half-Heusler) at room temperature. Low carrier concentration is taken in the simulation so that the mobility corresponds to the intrinsic value, while the experimental values are the highest reported ones (see compilation of these mobility values in Supplementary Table 4). The per-valley-density-of-state effective mass, obtained as $m_{DOS}^*/N_V^{2/3}$ where $m_{DOS}^*$ is the density-of-state effective mass and $N_V$ is the valley degeneracy, is a better estimation for transport effective mass which directly connects to the mobility. The shaded region illustrates the trend how mobility should vary as the transport effective mass if the deformation potential is kept a constant. **c** Predicted room-temperature power factors at optimal carrier concentration as functions of density-of-state effective mass. The power factor is approximately inversely proportional to the density-of-state effective mass. This trend is illustrated by the shaded area and is consistent with optical phonon limited transport (Supplementary Note 6 and Supplementary Fig. 16). **d**–**f** Calculated temperature-dependent power factor (PF), electrical conductivity ($\sigma$), and Seebeck coefficient ($S$) compared with the experiment for p-type NbFeSb. The carrier concentration in the simulation is taken to be the value (~2×$10^{20}$ cm$^{-3}$), which gives maximum power factor at room temperature

correspond to a (nearly) non-bonding behavior. This suggests the non-bonding signature characterized by vanishing COHP could potentially be used as an indicator for finding materials with weak electron–acoustic phonon coupling.

**Electron dynamics and transport.** The weak electron–acoustic phonon interaction as discovered translates into the low electron scattering rates, which sums up all scattering channels via EPI[23] (Methods). The channel (through acoustic phonons) that is normally expected to be largest, is now suppressed. As an example, ZrNiSn shows significantly smaller acoustic phonon scattering rates than the optical phonons (Fig. 3a and Supplementary Fig. 6), again confirming the strong suppression of electron–acoustic phonon coupling in the half-Heusler system.

With the scattering rates one can calculate the electron transport properties[31], for which we have also considered the effects of polar scattering and the screening due to free carriers, together with the electron-impurity scatterings (see Methods). Figure 3b shows the calculated mobility ($\mu$) compared with experimental values from different material families. Notably, half-Heuslers can exhibit intrinsic room-temperature mobility above 500 cm$^2$ V$^{-1}$ s$^{-1}$ despite their large density-of-state effective mass ($m^*$)—a direct consequence of their weak electron–phonon coupling. This example demonstrates how large mobility can benefit from weak EPI strength that can emerge from favorable orbital interactions. The combination of large density-of-state effective mass and weak EPI is particularly beneficial for thermoelectric materials, as suggested by the weighted mobility ($U = \mu m^{*3/2}$) frequently used as an indicator for large power factor. Figure 3c shows the power

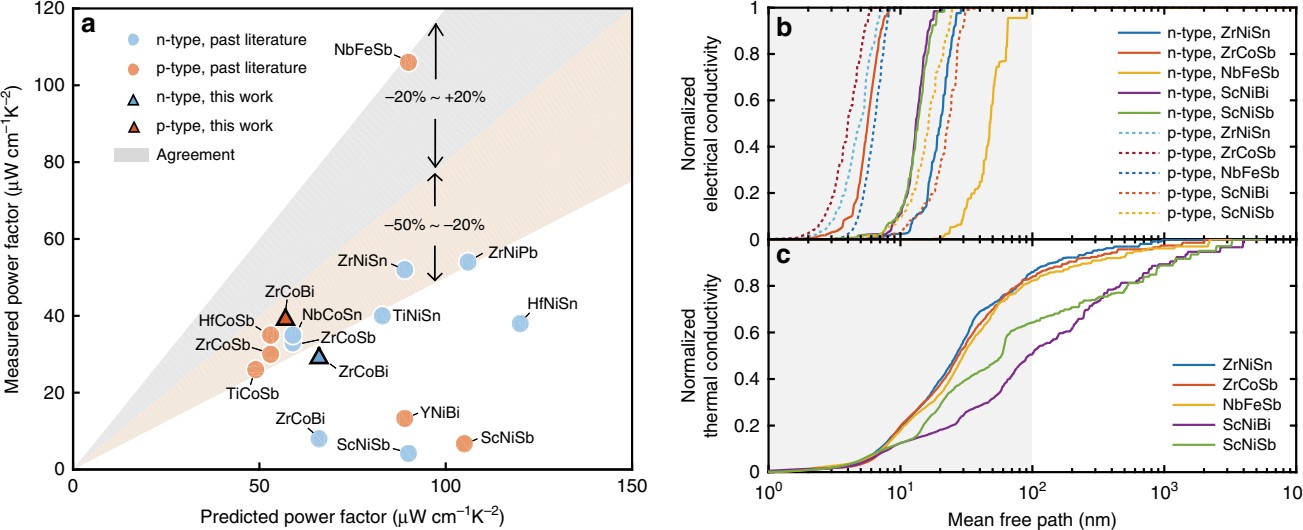

**Fig. 4** Evaluation of thermoelectric performance. **a** Power factors compared with experiments for various half-Heusler compounds. Only compositions with experimental values are shown here (not including alloys). The maximum power factors from experiments are taken to compare with the calculation (see more discussion on the temperature dependence in Supplementary Note 1, and the presented power factor values in Supplementary Tables 5 and 6). **b** The accumulated electrical conductivity with respect to the electron mean free path for both n- and p-type. **c** The accumulated thermal conductivity with respect to the phonon mean free path. The results are shown for materials with corresponding doping types that exhibit high-power factors according to Fig. 3c. The shaded area (<30 nm) denotes the MFP range where electrons mostly dominate the electrical transport but phonons contribute negligibly to the thermal conductivity

factor at optimal carrier concentrations in the half-Heusler compounds. Strikingly, several materials exhibit room temperature power factors higher than 100 $\mu$W cm$^{-1}$ K$^{-2}$ for both n- and p-type, and most of the compounds studied show power factors larger than 50 $\mu$W cm$^{-1}$ K$^{-2}$. We note that such large power factors are potentially achievable, as demonstrated by the recent example in p-type NbFeSb with a record-high-power factor of ~106 $\mu$W cm$^{-1}$ K$^{-2}$ at room temperature[20]. Figure 3d–f shows the calculated temperature-dependent electrical transport properties (conductivity, Seebeck coefficient, and power factor) for this compound. The good agreement compared with the experiment[20] justifies our computational framework, especially noting that there is no fitting parameter in our simulation.

## Discussion

We would also like to discuss the experimental aspect of realizing such large power factors and its consequence in the thermoelectric performance. Figure 4a compares the measured power factors with our predictions. Except for p-type NbFeSb, most other materials, however, have experimental power factors lower than the predicted values. In fact, to reach high-power factor in NbFeSb, the hot pressing temperature was significantly increased[20], leading to fewer defects and an intrinsic transport behavior—decreasing mobility as temperature increases, a signature rarely seen in half-Heuslers. Recent studies in ZrNiSn also indicate excess Ni creates electron scatterings that severely limit the charge transport[25], which is consistent with our mobility result for ZrNiSn (Supplementary Fig. 12). All these suggest that, if the defect concentrations can be reduced in half-Heuslers, one would expect higher power factors that line up better with our predictions. This has been corroborated by the preliminary results in our recently fabricated ZrCoBi samples (fabrication and characterization details in Methods), which demonstrated large power factors (for both n-type and p-type) reaching above 30 $\mu$W cm$^{-1}$ K$^{-2}$ (Supplementary Fig. 9), significantly improved over previous work[32] (8 $\mu$W cm$^{-1}$ K$^{-2}$ for n-type[2]) and closer to our predicted values (50~60 $\mu$W cm$^{-1}$ K$^{-2}$). The above discussion shows the prospect of reaching exceptionally high-power factors

>100 $\mu$W cm$^{-1}$ K$^{-2}$ for a variety of materials within the half-Heusler system at room temperature, by further optimizing the material processing and particularly controlling the defect concentrations. At higher temperatures, these theoretical power factors are expected to become smaller due to the stronger phonon scatterings. Still, power factors >70 $\mu$W cm$^{-1}$ K$^{-2}$ have been found in our calculations in select half-Heusler compounds (Supplementary Table 7), which hints at a big room for further improving the power factors of half-Heusler materials from room temperature to high temperatures.

The large power factors combined with reduced thermal conductivity through nanostructuring technique would lead to outstanding thermoelectric performance. To this end, we have performed thermal transport calculations for several compounds with large power factors. The accumulated contributions to the electrical and thermal conductivity (Fig. 4b, c) at room temperature show that electrons mostly have MFP's below 30 nm while phonon MFP's span a much wider range up to a few microns. This large disparity between the dominant electron and phonon MFP allows improved thermoelectric efficiency by reducing phonon thermal conductivity while maintaining the electron transport through grain boundary scattering[12,13,33]. At higher temperatures, the dominant electrons' MFP becomes even smaller, on the order of 10 nm (Supplementary Fig. 7 and Supplementary Table 7, calculation at 1000 K). We note that the grain sizes in up-to-date half-Heusler nanocomposites are around hundreds of nanometers[34]. If the grain sizes can be pushed down to tens of nanometers, one can potentially achieve even larger thermoelectric efficiency in these half-Heusler compounds, particularly benefiting from their exceptionally large power factors.

In summary, through the first principles study of EPIs, we have revealed that the origin of the remarkably high-power factors in half-Heusler materials lies in suppressed electron–acoustic phonon couplings. These weak couplings, exemplified by low deformation potentials, emerge from crystal symmetry-protected nonbonding orbitals at the band edge. The vanishing bonding (antibonding) orbital interactions make the half-Heuslers unique material platforms that bypass the traditional viewpoint

according to which they should have acoustic-phonon-limited electron transport. The understanding of these couplings provides a strategy for a material to have simultaneously large density-of states (beneficial for a large power factor, but often associated with low mobility) and high electron mobility. Our finding foresees that there is significant room to improve the thermoelectric performance of half-Heusler systems. These new insights will stimulate not only the discovery of novel high-performance thermoelectric materials, but also the development of high-mobility materials for microelectronic and optoelectronic applications.

## Methods

**Material selection**. A half-Heusler material consists of three atoms in the unit cell (denoted as ABC). The crystal structure of half-Heusler materials has only one parameter—the lattice constant $a$. Given this number, the positions of the three atoms (denoted as ABC, where B occupies the tetrahedral site AB forms the zincblende structure while C is the main group element, for example in ZrNiSn) in the unit cell can be labeled by $r_A = (0,0,0)a$, $r_B = (1/4,1/4,1/4)a$, and $r_C = (1/2, 1/2,1/2)a$, respectively (atomic arrangement as given in the inset of Fig. 1a, b). The selection of half-Heuslers for our calculation is based on a recent material genome search[21], which examines all stable half-Heusler compounds for B=Co or Ni, as also confirmed by their experimental studies. From the list, we select all compounds excluding those with rare elements (Ru, Os, Rh, Ir, Pd, Pt, Ag, and Au), adding up to 14 compositions in total. Half-Heuslers with other element types for B were not studied in this material search, but are known to have good power factor from experiments. One example is NbFeSb (B=Fe), which possesses highest reported power factor in semiconductors above room temperature and has also been included in our calculation[20]. Therefore a total number of 15 half-Heusler compounds are examined. This selection does not present any bias towards particular composition.

**Electronic structure calculation**. The equilibrium properties of electrons are calculated from first principles using the QUANTUM ESPRESSO package[35]. We use the generalized gradient approximation (GGA) of Perdew, Burke, and Ernzerhof[36] with the Trouiller-Martins-type norm-conserving semilocal pseudopotential (corresponding to pbe-mt.UPF in the QUANTUM ESPRESSO pseudopotential library). A cutoff energy of 120 Rydberg and a $6 \times 6 \times 6$ k-mesh are used to determine the equilibrium lattice constant for half-Heusler materials, as given in Supplementary Table 1. These are also the same parameters used to calculate the band structure (Fig. 2a, c)and to feed into the electron–phonon interpolation procedure (require band energies and electronic wavefunctions).

Half-Heusler materials involve heavy elements, and spin-orbit couplings (SOC) could potentially affect the band shapes near the band edge and thus alter the electron transport. For this we have checked the band structures calculated with and without SOC (Supplementary Fig. 10). The results indicate that the SOC only has negligible effects on the band shapes near the band edges. To facilitate the computation, the results reported are, therefore, obtained without including the SOC effect. We also note that these materials are not magnetic, as the electronic structure calculation by starting from non-zero atomic magnetizations have eventually converged to a state with zero magnetization, which indicates that non-magnetic state is the ground state of the system. The total energies between non-zero and zero starting magnetizations also lead to almost the same values (Supplementary Table 8). In the transport calculation, we have thus assumed non-spin polarized configuration.

**Deformation potential**. For acoustic phonons it can be shown that at the long wavelength limit the electron–phonon coupling matrix vanishes with the phonon wave vector $q$ in a linear fashion:

$$\left| \langle \psi_k | \partial_q V | \psi_{k+q} \rangle \right| \approx \Xi q \quad (1)$$

The proportionality factor $\Xi$ is defined as the acoustic deformation potential. We note that such a deformation potential is uniquely defined based on the EPI, and directly relates to the electron transport. An alternative definition regards the deformation potential as the band energy change when the lattice is strained (as originally suggested by Bardeen and Shockley[26]), and often used to study band alignment at interfaces. However, this definition suffers from the problem of ill-defined energy reference in the first principles framework, and therefore not employed in our work.

For non-polar optical phonons, the electron–phonon coupling matrix in general does not vanish at the long wavelength limit. However, based on the group theory[37], the coupling matrix for nondegenerate electron states is zero if the representations of the optical phonon do not contain identity representation. For half-Heusler materials, the optical phonons at $q = 0$ are threefold degenerate (neglecting the long-range effect, i.e., the LO-TO splitting), corresponding to the $T_2$ representation of group $T_d$. At the conduction band edge at the X-point with group

$D_{2d}$, $T_2$ reduces to $B_2 + E$. The decomposed representations do not contain identity representation, and therefore the electron–optical phonon coupling vanishes at $q = 0$ for EPI matrix with initial electron at the X-point. For small wave vectors we found that a linear behavior fits well for the coupling matrix, and in similar fashion as acoustic phonons we define the deformation potential for non-polar optical phonons (in unit of energy), which characterizes the strength of electron–optical phonon scattering, as shown in Fig. 1d.

**Symmetry analysis**. The symmetry analysis based on group theory for analyzing the interaction between atomic orbitals and their formation into molecular (or crystal, in the case of solids) orbitals can categorize the atomic orbitals into different symmetry representations[27]. One first determines the symmetry group associated with the crystal structure. For the electron energy at a given k point, we then reduce the crystal symmetry group to the so-called small group of k (the collection of the symmetry operations in the crystal symmetry group that do not alter the wave vector k), which governs the orbital interactions at k. For X-point the small group is $D_{2d}$ while for L-point it is $C_{3v}$.

In a tight-binding model (or LCAOs), one writes the crystal wavefunction as superpositions of atomic orbitals:

$$\psi = \sum_{i\alpha} c_{i\alpha} \varphi_{i\alpha} \quad (2)$$

where $\varphi$ represents each atomic orbital $\phi$ summed into its Bloch form (for our study, the "atomic orbital" is understood as in its Bloch summation form) and the indices $i$ and $\alpha$ denote different atomic sites and orbital types, respectively.

$$\varphi_{i\alpha}(r) = \frac{1}{\sqrt{N}} \sum_{R_i} e^{ik \cdot R_i} \phi_{i\alpha}(r - R_i) \quad (3)$$

To obtain the band structure, one usually solves the eigenvalue problem (obtain the prefactor $c_{i\alpha}$), which relies on the interaction between different atomic orbitals through the Hamiltonian $\hat{H}$ - $\langle \varphi_{i\alpha} | \hat{H} | \varphi_{j\beta} \rangle$. The application of group theory is to find certain coupling matrix that is zero when the atomic orbitals belong to different representations. To characterize the representations particularly for the crystal orbitals as given by Eq. 3, we note that they are in the form of product of two terms, with one being the phase factor $e^{ik \cdot R_i}$ and the other being the true atomic orbital $\phi_{i\alpha}$ on a single atomic site. The representation is, therefore, described by the product representation of these two, which can be derived from group theory. Detailed derivation can be found in Supplementary Note 4 and Supplementary Table 9–11.

**Crystal orbital overlap population**. The crystal orbital overlap population (COHP) was developed to visualize energy-resolved chemical bonding for solids, by partitioning the band structure energy into contributions from each orbital pair[30]. Following the tight-binding model as given in Eqs. 2 and 3, the COHP can be defined as:

$$\text{COHP} = \sum_{i\alpha, j\beta} c_{i\alpha}^* c_{j\beta} \langle \varphi_{i\alpha} | \hat{H} | \varphi_{j\beta} \rangle \quad (4)$$

The product $c_{i\alpha}^* c_{j\beta}$ characterizes whether the orbitals are in phase or not, indicating their bonding character. The Hamiltonian matrix element further takes into account the differences in the bond strength. This is important because the deformation potential depends on the energetic scales and s–s with greater bonds (therefore larger overlap) should behave differently than p–p interactions. To evaluate COHP's based on first principles calculations, one needs to construct an effective tight-binding model. This is achieved by projecting the wavefunctions onto the local atomic orbitals. In this sense the COHP's as calculated should be more strictly referred to as pCOHP (p means "projected"). More details in the numerical implementation of COHP and its derivation can be found in Supplementary Note 5.

**Thermoelectric transport calculation**. Here, we elaborate procedures to calculate the thermoelectric transport properties from first principles. For electrons at normal temperatures, their intrinsic scattering rates (or inverse of relaxation times) are governed by EPIs, and can be derived based on Fermi's Golden rule[31],

$$\frac{1}{\tau_{kn}} = \frac{\pi}{m_0 N_q} \sum_{q,m,q\lambda} \frac{1}{\omega_{q\lambda}} \left| \langle \psi_{kn} | \partial_{q\lambda} V | \psi_{k+qm} \rangle \right|^2$$
$$\times \left[ \left( n_{q\lambda} + f_{k+qm} \right) \delta \left( E_{kn} - E_{k+qm} + \hbar\omega_{q\lambda} \right) \right.$$
$$\left. + \left( n_{q\lambda} + 1 - f_{k+qm} \right) \delta \left( E_{kn} - E_{k+qm} - \hbar\omega_{q\lambda} \right) \right] \quad (5)$$

where the band indices (n, m, λ) are specified for the electron–phonon coupling matrix, $m_0$ is the unit cell mass, $N_q$ is the number of points for the phonon mesh, $E$ is the electron energy, $\omega$ is the phonon frequency, $f$ is the Fermi-Dirac distribution function for electrons and $n$ is the Bose-Einstein distribution function for phonons. Direct first principles simulation of electron transport has been formidable in the past due to the large phase space involved in the scattering rate calculation, which

requires the knowledge of EPI on very fine electron and phonon meshes to reach the convergence. This becomes a viable option only recently owing to the development of Wannier function-based interpolation scheme for EPI[23]. We have used the EPW package[38] to interpolate electron–phonon coupling matrices, based on the electronic information obtained as explained above, as well as the phonon information calculated on a $6 \times 6 \times 6$ **q**-mesh (including phonon dispersion and perturbed potential - $\partial_{\mathbf{q}\lambda} V$ in the coupling matrix). The quantities needed for transport calculations—electron energies, phonon frequencies, and electron–phonon couplings—are then mapped to much finer meshes through the real-space Wannier functions. We have checked that the mapped quantities agree well with the directly calculated values from DFT (Supplementary Fig. 11). The fine meshes we used for the transport property calculations are given in Supplementary Table 2. This method readily leads to EPI on fine meshes without incurring much computational burden, thereby largely accelerating the calculation. The electron scattering rates are calculated by summing over all possible scattering channels based on a tetrahedral integration method[39].

Given the electron relaxation times, the electron transport properties can be derived from the Boltzmann transport equation[31]:

$$\sigma = \frac{e^2}{3\Omega N_{\mathbf{k}}} \sum_{\mathbf{k}n} \mathbf{v}_{\mathbf{k}n}^2 \tau_{\mathbf{k}n} \left( -\frac{\partial f_{\mathbf{k}n}}{\partial E} \right) \tag{6}$$

$$S = \frac{e}{3\sigma\Omega N_{\mathbf{k}} T} \sum_{\mathbf{k}n} (E_{\mathbf{k}n} - \mu) \mathbf{v}_{\mathbf{k}n}^2 \tau_{\mathbf{k}n} \left( -\frac{\partial f_{\mathbf{k}n}}{\partial E} \right) \tag{7}$$

where $e$ is the electronic charge, $\Omega$ is the unit cell volume, $N_{\mathbf{k}}$ is the number of points for the electron mesh, $\mathbf{v}_{\mathbf{k}n}$ is the group velocity of electron, and $\mu$ is the Fermi level. With these, the power factor $\sigma S^2$ is readily calculated. A prototypical temperature dependence of the electron mobility is given in Supplementary Fig. 13, where we show that it is not convincing enough to conclude whether the electron scattering is acoustic phonon or optical phonon dominated by just examining the temperature dependence. For a more realistic estimation of the transport properties (e.g., mobility and power factor), we have added the effects due to polar interaction[40,41] and the carrier screening based on analytic models (Supplementary Note 2). The electron-impurity scattering is also considered using the Brooks–Herring model[42] (Supplementary Note 1), which considers the long-range Coulomb field induced by the ionized impurity, but neglects the short range interactions due to distortions around the impurity. We also note that the impurity scattering can also cause the optimal power factor to show a peak with respect to the temperature increase (Supplementary Fig. 14), which is often observed in experiments.

For thermal conductivity calculations, we first calculate phonon relaxation times based on the three-phonon scattering processes, which are characterized by the anharmonic force constants (we restrict to third-order force constants in our study). The harmonic and third-order force constants are fitted together based on first principles calculations that yield forces acting on different atoms for different sets of displacements in a supercell ($2 \times 2 \times 2$ conventional unit cells, 96 atoms). The harmonic force constants provide the phonon dispersion while the third-order force constants are used to calculate the phonon relaxation times $\tau_{\mathbf{q}}\lambda$. Given these, the thermal conductivity is obtained as

$$\kappa_{ph} = \frac{1}{3\Omega N_{\mathbf{q}}} \sum_{\mathbf{q}\lambda} \mathbf{v}_{\mathbf{q}\lambda}^2 \tau_{\mathbf{q}\lambda} \hbar\omega_{\mathbf{q}\lambda} \frac{\partial n_{\mathbf{q}\lambda}}{\partial T} \tag{8}$$

where $\hbar$ is the reduced Planck constant and $\mathbf{v}_{\mathbf{q}\lambda}$ is the phonon group velocity. For thermal conductivity we have neglected the alloying or impurity scattering effects, because they generally depend on the composition and vary from case to case. Nonetheless, inclusion of these effects lowers the thermal conductivity, which will make the thermoelectric efficiency even higher. We have compared our results with past first principles simulations[43] and the thermal conductivity values are comparable (Supplementary Table 3). More calculation details can be found in Supplementary Note 1.

**Material fabrication and characterization**. We prepared the n-type $ZrCo_{0.9}Ni_{0.1}Bi$ and p-type $ZrCoBi_{0.8}Sn_{0.2}$ samples using ball-milling method. Pure elements (Zr 99.2%, Co 99.8%, Bi 99.999%, Sn 99.8%, and Ni 99.7%, Alfa Aesar) were loaded in a stainless steel jar according to the stoichiometry and ball milled (SPEX 8000 M Mixer/Mill) for 20 h. The ball milled powder were then compressed to disk shape by a direct current hot pressing process at about 900 °C for 5 min under 50 MPa.

Phase identification of the samples (Supplementary Fig. 8) were carried out by X-ray diffraction (XRD) on a PANalytical multipurpose diffractometer with an X'Celerator detector (PANalytical X'Pert Pro). The electrical conductivity and Seebeck coefficient were measured using a commercial (ZEM-3, ULVAC) system under a helium atmosphere at varying temperatures (Supplementary Fig. 9). The power factors are then calculated based on these measured data, with the maximal

values for the n-type and p-type samples reaching ~30 μW cm$^{-1}$ K$^{-2}$ and ~40 μW cm$^{-1}$ K$^{-2}$, respectively.

**Data availability**. The data that support the findings of this study are available from the corresponding authors on reasonable request.

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

## Acknowledgements
We thank Weishu Liu, Lei Sun, Bing Yan, Qiong Ma, Suyang Xu, Zhiwei Ding, and Mingda Li, for the inspiring discussions. This article is based upon work supported as part of the Solid-State Solar-Thermal Energy Conversion Center (S³TEC), an Energy Frontier Research Center funded by the U.S. Department of Energy (DOE), Office of Science, Basic Energy Sciences (BES), under Award # SC0001299/DE-FG02-09ER46577 (for fundamental research on electron–phonon interaction in thermoelectric materials) and by the DARPA MATRIX program, under Grant HR0011-16-2-0041 (for code development to support practical thermoelectric devices).

## Author contributions
J.Z., T.-H.L., and Q.S. carried out the first principles calculations. H.Z., R.H, J.M., Z.L., and W.R. fabricated and characterized the thermoelectric samples. J.Z. and G.C. analyzed the data and wrote the manuscript. Z.R. pushed for the theoretical understanding of the high-power factor. G.C. supervised the research. All authors commented on, discussed and edited the manuscript.

## Additional information

**Competing interests:** The authors declare no competing interests.

