## [Peer Review File · Nature Communications]

Reviewers' Comments:

Reviewer #1:

Remarks to the Author:

Half-Heusler compounds have attracted attentions to the thermoelectric community as a well-known high power factors system, which bearing a cubic crystal structure with three atoms per unit cell. However, the origin of these high power factors was not clear. Here, the authors have pointed out the strong suppression of electron-acoustic phonon couplings enhanced the power factors of Half-Heusler materials, which is contradicts the sense that the acoustic phonon scattering limit the electron transport. More accurate Wannier interpolation scheme for electron-phonon interaction was adopted in this work. It shows that the vanishing bonding character can be protected by the crystal symmetry. The work comprises substantial novelty and significance. I believe it is important to publish the present results as this paper provides new insight to understand the origin of high power factors in Half-Heusler materials. I would recommend accepting this paper in Nature communications after minor revision. These are the few remarks which author should explain during revision.

Comment 1: A part of caption for Figure 3 in manuscript was lost, the authors should make an adjustment in the main text.

Comment 2: Authors selected 15 half-Heusler compounds, and some of them include Sn, Sb, Bi elements. It is well known that the spin-orbital coupling (SOC) of these elements is significant, but the authors do not mention or/and check it throughout the manuscript and supplementary information. It is better to consider the SOC effect for these heavy elements.

Reviewer #3:

Remarks to the Author:

The authors present combined theory/experiment study of electron phonon interactions in half-Heusler compounds. The main finding of the paper is the explanation of large power factors found in this class of systems. According to the authors large power factors are due to relatively weak electron-acoustic phonon interactions as compared to the interactions between electrons and optical phonons. This is important as, according to the authors, the findings enhance significantly the present thinking about electron-phonon interactions in half-Heuslers. The explanation for the weaker electron-acoustic phonon interactions (than previously thought) the authors find in the crystal symmetry and the nature of electronic states close to the band edges. In my view, the results are very interesting and novel and could be relevant if correct. However, I have a number of comments listed below and cannot recommend this manuscript for publication before they are appropriately addressed.

1) First, it is unclear whether the calculated electron-phonon coupling and corresponding deformation potentials shown in Figure 1 are for the valence or conduction bands? I think both need to be shown and the differences and/or similarities need to be discussed.

2) The explanation of the observed trends from Fig. 1 is fairly qualitative and needs to be quantified. Determining the bonding, anti-bonding or non-bonding nature of electronic states in solids is highly non trivial and the authors need to do a better job in convincing the readers that what they are suggesting is indeed true. Presently, we need to blindly trust the authors and rely on some undefined criteria. Their symmetry analysis is questionable as discussed in the next comment.

Moreover the statement at the beginning of page 6: "While this state (representation E) no longer derives from a single atomic orbital, its bonding (anti-bonding) interactions with nearby atomic

orbitals with higher (lower) energies make itself mostly consist of d orbitals from site B (Fig. 2f), as confirmed by the projected density of states (pDOS, Fig. 2b). Due to the cancellation of bonding and anti-bonding interactions, this state has energy close to that of a single atomic orbital, closely resembling a non-bonding orbital and therefore is expected to also have small deformation potential." is very qualitative and very troublesome. Without some kind of quantification I have hard time understanding how do bonding and anti-bonding interactions cancel and why would a state that interacts with neighboring atoms be expected to have small deformation potential? If the reason is the d-character, does this mean that the d-orbitals will always have small deformation potentials?

3) Also, I disagree with the molecular orbital diagrams shown in Fig. 2. Why don't $d_{x^2-y^2}$ orbitals from atom A and atom B interact? There is no symmetry constraint preventing them from interacting and d-orbitals of Nb are not that confined? The possibility of this interaction conflicts with the explanation offered by the authors.

4) Even if we assume that $d_{x^2-y^2}$ orbitals from A and B sites do not interact, the symmetry constraints apply only to the CBM of NbFeSb and it is unclear from the paper whether the VBM formed of d_{xz} and d_{yz} orbitals from the site B, and p_x and p_y from the site C would exhibit similarly weak electron-acoustic phonon interactions or not? Experimental results from Fig. 3 are for the p-type NbFeSb, hence, pertinent to the VBM and not CBM! Analogous question applies to the CBM of ZrNiSn.

5) I have hard time imagining that there are no localized spins on Fe or Ni atoms in these systems and from what I can tell non-spin polarized calculations have been performed. Including spin degrees of freedom can significantly affect the electronic band structures and phonon dispersions and could influence the results. I cannot see in the paper what would be the effect of including spins?

6) Also, I am a big proponent of the reproducibility of results and am very much against putting all the technical details in the supplementary materials. The paper needs to be self-contained and the minimal info that allows reproducing results needs to be provided in the main text. Presently, that is not the case, calculations are referred to as first-principles without any further information on what that actually means. And as important, by putting all the info about the tools in the supplementary section the authors are not giving proper credit to the developers whose tools are being used to obtain the results. This is simply wrong.

Reviewer #4:

In the past years, most efforts on half-Heusler TE materials were focused on suppressing the phonon transport, relatively less attention had been paid for the intrinsic reasons leading to their high power factors. Beyond the high band degeneracy that benefits the Seebeck coefficient, low deformation potentials in typical half-Heusler TE materials, such as (Zr,Hf)NiSn, (V,Nb)FeSb, have also been experimentally justified contributing to their relatively high carrier mobilities. However, the intrinsic origin of the low deformation potential is still elusive. The manuscript by Zhou et al. revealed that the origin of low deformation potential in half-Heuslers lies in the weakened electron-acoustic phonon couplings, which further emerge from crystal symmetry-protected non-bonding orbitals. This is a new insight that might be helpful to design new TE materials with enhanced power factor. There are some comments that should be considered by the authors:

- 1) Generally, the conduction band CB and valence band VB of half-Heuslers are dominated by d orbitals, which is more localized than s or p orbitals, and thus more difficult to "deform". My question is whether the non-bonding orbital can only be found in d orbital dominated system or can it also exist in the s or p orbitals dominated system. As the CB and VB of most good thermoelectrics are dominated by s and p orbitals. More discussion on this point would improve the quality of the manuscript.

- 2) In Figure 3a, the electron scattering rate of ZrNiSn was calculated with a carrier concentration of 10^{18} cm^{-3} . However, the general optimal carrier concentration of Half-Heusler TE materials is in the range of $10^{20} - 10^{21} \text{ cm}^{-3}$. Besides, the optical phonon usually dominates the carrier transport at low carrier concentration, while at higher carrier concentration, the acoustic phonon contributes more (Ref. 25). Adding the electron scattering rate at higher carrier concentration ($10^{20} - 10^{21} \text{ cm}^{-3}$) is highly suggested to identify the dominated carrier transport mechanism in actual half-Heusler TE system.
- 3) In Figure 3c, the authors calculated the optimal power factor at room temperature. While half-Heuslers typically show good TE performance at higher temperature $> 800 \text{ K}$, it is suggested to add the result at higher temperature.
- 4) The unit of power factor in the final part of the main text is wrongly written.

Reviewer #1:

General comment: Half-Heusler compounds have attracted attentions to the thermoelectric community as a well-known high power factors system, which bearing a cubic crystal structure with three atoms per unit cell. However, the origin of these high power factors was not clear. Here, the authors have pointed out the strong suppression of electron-acoustic phonon couplings enhanced the power factors of Half-Heusler materials, which is contradicts the sense that the acoustic phonon scattering limit the electron transport. More accurate Wannier interpolation scheme for electron-phonon interaction was adopted in this work. It shows that the vanishing bonding character can be protected by the crystal symmetry. The work comprises substantial novelty and significance. I believe it is important to publish the present results as this paper provides new insight to understand the origin of high power factors in Half-Heusler materials. I would recommend accepting this paper in Nature communications after minor revision. These are the few remarks which author should explain during revision.

Response: We are glad that the reviewer finds our work appealing and we thank the reviewer for his/her comments. The reviewer's main suggestion is to add discussion on the effect of spin-orbit coupling (SOC) on our electron transport modeling. For this, we have added comparison of the band structures calculated with and without SOC effect. The results show very small changes between these two, and we thus believe the calculation results without including the SOC effect can be justified. We have added corresponding discussions on this point in the manuscript.

Comment 1: A part of caption for Figure 3 in manuscript was lost, the authors should make an adjustment in the main text.

Response and Revision: We thank the reviewer for pointing this out. This happens when we convert our word file into pdf. We have accordingly adjusted the figure so that pdf file now contains the full caption.

Comment 2: Authors selected 15 half-Heusler compounds, and some of them include Sn, Sb, Bi elements. It is well known that the spin-orbital coupling (SOC) of these elements is significant, but the authors do not mention or/and check it throughout the manuscript and supplementary information. It is better to consider the SOC effect for these heavy elements.

Response: We have carefully checked the band structures, and have found that half-Heusler materials, different from some narrow-gap thermoelectric materials, are barely affected by the SOC effect. Below we show the band structure with and without the SOC effect for both NbFeSb and ZrNiSn, near their band edges (Fig. R1). It can be seen that the SOC barely changes the band shape (therefore effective mass and other transport properties should remain the same). We note that the splitting of energy bands at X point in the valence band of ZrNiSn has small effects on the transport, because the energy level at X is ~ 0.3 eV away from the band edge, which is one order of magnitude larger than typical thermal excitation energy ($k_B T \sim 0.025$ eV at 300K). The electrons at X point thus contribute very little to the total transport. In sum, we think our calculation results based on band structures without SOC effect are justified.

Figure R1. Band structures near the band edge for (a) NbFeSb, and (b) ZrNiSn, with and without the spin-orbit couplings (SOC). The results without SOC are obtained using PAW pseudopotentials without relativistic corrections, while the results including SOC are obtained using fully-relativistic PAW pseudopotentials [1].

Revision: To comment on the SOC effect, we have added a paragraph in Methods section, and have put the figure into the supplementary information.

Reviewer #3:

General comment: The authors present combined theory/experiment study of electron phonon interactions in half-Heusler compounds. The main finding of the paper is the explanation of large power factors found in this class of systems. According to the authors large power factors are due to relatively weak electron-acoustic phonon interactions as compared to the interactions between electrons and optical phonons. This is important as, according to the authors, the findings enhance significantly the present thinking about electron-phonon interactions in half-Heuslers. The explanation for the weaker electron-acoustic phonon interactions (than previously thought) the authors find in the crystal symmetry and the nature of electronic states close to the band edges. In my view, the results are very interesting and novel and could be relevant if correct. However, I have a number of comments listed below and cannot recommend this manuscript for publication before they are appropriately addressed.

Response: First of all, we want to thank the reviewer for finding our work interesting and we appreciate his/her time in providing detailed comments particularly on our analysis of crystal symmetry and the orbital interactions. We note that the reviewer's major comments are on how to quantify the bonding and anti-bonding interactions, and how these can lead to small deformation potentials for a band edge state, especially for VBM of NbFeSb and CBM of ZrNiSn. The reviewer also raised a question regarding the correctness of our symmetry analysis. Here we give a short response to these questions, while detailed explanations will be followed.

In general, our statement on the link between orbital interactions and deformation potentials is built upon three carefully defined studies: 1) symmetry analysis of orbitals; 2) projected density of states; 3) quantitative crystal orbital Hamilton analysis compared to deformation potentials. In this revised manuscript, we have added a fourth part – 4) a simplified three-level tight-binding model, to further illustrate how weakly bonding or anti-bonding states can lead to a small deformation potentials.

We want to start from the symmetry analysis, which is the basis of our work. Our symmetry analysis is built upon crystal orbitals, which are often used as the basis sets in tight-binding models to solve band structure of crystals. They are linear combinations of atomic orbitals, and are what we refer to in the orbital diagram (Fig. 2e-f). We believe the reviewer's conclusion on the correctness of our symmetry analysis is probably misled by our drawing which seems to suggest those orbitals to be atomic orbitals localized on single atoms (instead of crystal orbitals). We have clarified such difference in the following as well as in the manuscript. We also mention, the correctness of our symmetry analysis is confirmed by comparing our theory with the symmetry labels directly outputted from the DFT calculations. The non-bonding orbital in NbFeSb is a direct result of such symmetry protection (Fig. 2e).

The next question raised by the reviewer is how we should understand the valence band edge state in NbFeSb as well as conduction band edge of ZrNiSn, because these two do not represent a single non-bonding state, but emerge from the interactions among many orbitals. For these, we introduce the idea of nearly non-bonding orbital (Fig. 2f), which represents the state whose interactions with other levels compete in such a way that it is a weakly bonding or weakly anti-bonding level (we have called it a “cancellation of bonding and anti-bonding interactions” but feel this might have

led to some confusions as the reviewer pointed out). In this scenario, we have combined our orbital diagram with projected density of states to conclude that these states are indeed weakly bonding or anti-bonding.

To further establish the connection, in the revised manuscript, we have built a three-level system to illustrate how a weakly bonding or anti-bonding state would have a small deformation potential. In this three-level model, the weakly bonding level results from an intermediate state that interacts with higher (lower) energy level which gives to bonding (anti-bonding) character to itself. It is the competition (or, cancellation) between the bonding and anti-bonding interactions that leads to this weakly bonding state, with eventually a small deformation potential. This model, together with our analysis above, more quantitatively explains how a nearly non-bonding state in the VBM of NbFeSb and CBM of ZrNiSn should have small deformation potentials.

Furthermore, the weak bonding characters have been checked through the quantitative crystal orbital Hamilton analysis, and the results indicate that they are indeed weakly bonded. We have also provided deformation potential results for the valence band states for all fifteen half-Heusler compounds to show that the acoustic phonon deformation potentials are indeed much weaker in the valence band as well. Detailed responses to other comments are given below.

Comment 1: First, it is unclear whether the calculated electron-phonon coupling and corresponding deformation potentials shown in Figure 1 are for the valence or conduction bands? I think both need to be shown and the differences and/or similarities need to be discussed.

Response: In Figure 1 we only showed deformation potentials for conduction bands. We have also checked valence bands but did not plot them because valence bands near the edge are degenerate, meaning that corresponding electron-phonon coupling matrices involve DFT wavefunctions that are arbitrary superpositions of degenerate eigenfunctions, making direct extraction of deformation potential from the electron-phonon couplings complicated. In order to present the electron-phonon couplings also for the valence band, we define an “averaged” electron-phonon coupling matrix by averaging over degenerate electronic states (based on Sjakste *et al*'s work [2], formula 4, although we note this is up to one's definition), where N_d is the degeneracy number, and n (m) denotes the band index that sums over the degenerate states at point \mathbf{k} .

$$\bar{D}(\mathbf{k}, \mathbf{k} + \mathbf{q}) = \sqrt{\sum_{n,m=1}^{N_d} |D_{nm}(\mathbf{k}, \mathbf{k} + \mathbf{q})|^2 / N_d^2} \quad (\text{R1})$$

This allows us to examine the wave vector dependence of coupling matrix and thus to extract an effective deformation potential (Fig. R2). We note that one difference in electron-optical-phonon couplings between conduction and valence bands is that, in the conduction band, the coupling matrix goes to zero as phonon wave vector approaches zero (dictated by symmetry, as we explained in Methods section), while in valence band, there is no symmetry to constrain its value and thus for optical phonons the coupling matrix is nearly a constant as normally expected [3]. It can be seen that coupling matrices associated with optical phonons are generally much larger than those with acoustic phonons (Fig. R2a-b).

Noting that optical phonon coupling matrix is almost a constant, a direct fit would give a constant, which however has unit [energy / length]. To have a more straightforward comparison with acoustic deformation potential, we divide this constant by a wave vector in unit of [$1 / \text{length}$], which is one fifth of the Brillouin zone size, characteristic of the typical length of phonon wave vector that can interact with thermally excited electrons ($\sim 5k_B T$). The obtained optical phonon deformation potentials as well as acoustic ones are compared in Fig. R2c-d for all the materials we have examined. Generally, in valence bands acoustic phonon deformation potentials are much smaller than optical phonon deformation potentials, which is consistent with our results in Fig. 1c-d. Compared to the conduction band (Fig. 1), we note that the approximate order-of-magnitude for acoustic and optical deformation potentials do not exhibit significant difference – the acoustic phonon deformation potentials are around 0.5 – 1 eV, while optical ones are mostly within 2.5 – 4 eV.

Figure R2. (a)-(b) Electron-phonon coupling matrix along high symmetry lines for valence band edge states. For NbFeSb and ZrCoSb the initial electron state is located at L point while for ZrNiSn it is at Γ point. (c)-(d) Averaged deformation potential for acoustic and optical phonons.

Revision: We have mentioned the valence band results regarding the comparison between acoustic phonon deformation potential and optical ones in the manuscript. The above figure has been added into the supplementary information, and the discussion above is added into its figure caption.

Comment 2: The explanation of the observed trends from Fig. 1 is fairly qualitative and needs to be quantified. Determining the bonding, anti-bonding or non-bonding nature of electronic states in solids is highly non trivial and the authors need to do a better job in convincing the readers that what they are suggesting is indeed true. Presently, we need to blindly trust the authors and rely on some undefined criteria. Their symmetry analysis is questionable as discussed in the next comment.

Response: We appreciate the reviewer's comments and agree that determining the bonding character of electronic states in solids can be a non-trivial task. However, we believe our results are drawn upon carefully defined criteria. First, we understand that bonding analysis can be complicated by multiple orbital interactions among different orbital types. However, the symmetry analysis is rigorous regardless of the detailed interactions (we will clarify our symmetry analysis in more detail in the response below). As a result, the existence of non-bonding state is guaranteed by the crystal symmetry, and in our case, it can appear at the band edge, which contributes to electron transport. Second, if the orbital is non-bonding, we expect projected density-of-states will consist majorly of one orbital type. This expectation is confirmed by our pDOS plots as shown in Fig. 2(b,d), which is another evidence.

With symmetry and pDOS analysis, we have also used another approach to seek a quantitative link between bonding character and the deformation potential. Quantifying bonding character in molecular orbitals is traditionally proceeded via Mulliken population analysis [4], [5] for molecular orbitals. The generalization of this idea to periodic solids is pioneered by Hoffman [6], [7] and extended by Dronskowski and Bloechl [8] that lead to the concept of crystal orbital Hamilton population (COHP) analysis. We have calculated the COHP for conduction band edge states and compared them with corresponding acoustic deformation potentials (see Fig. S2, and Supplementary Note 5). It can be seen that not only the nearly non-bonding feature is captured by the close-to-zero COHP values, the order-of-magnitude is also comparable. In particular, it seems from our analysis the deformation potential values are bounded by the COHP values. We believe our comparison between COHP and deformation potential further provides a quantitative evidence of the connection between these two. We also acknowledge that in terms of the exact number the COHP can still differ from some other measures of the bonding character, but our main message here is that the nearly non-bonding character will mostly lead to a small deformation potential, which will not be altered if one resorts to a different measure of the bonding character.

Moreover, we have also built a three-level system to clearly show how a low deformation potential can emerge from a state interacting with two other levels that have mixed bonding and anti-bonding interactions (as given in our next response). We hope the above discussions, combined with the three-level illustration, will mostly answer the reviewer's question.

Revision: We have expanded and re-written the **crystal orbital analysis** section in the manuscript (starting from the second paragraph in that section). We emphasized that the symmetry analysis is general and does not depend on the interaction details. We also explained the picture of non-

bonding orbital with more details given by the three-level system. A new section in the supplementary information (Supplementary Note 3) is written to elaborate on the three-level system.

Moreover the statement at the beginning of page 6: “While this state (representation E) no longer derives from a single atomic orbital, its bonding (anti-bonding) interactions with nearby atomic orbitals with higher (lower) energies make itself mostly consist of d orbitals from site B (Fig. 2f), as confirmed by the projected density of states (pDOS, Fig. 2b). Due to the cancellation of bonding and anti-bonding interactions, this state has energy close to that of a single atomic orbital, closely resembling a non- bonding orbital and therefore is expected to also have small deformation potential.” is very qualitative and very troublesome. Without some kind of quantification I have hard time understanding how do bonding and anti-bonding interactions cancel and why would a state that interacts with neighboring atoms be expected to have small deformation potential? If the reason is the d-character, does this mean that the d-orbitals will always have small deformation potentials?

Response: We want to use a tight-binding model to illustrate the idea of cancellation of bonding and anti-bonding interactions, and its connection to small deformation potentials. For simplicity, we consider three levels, each of which comes from one atom with different orbital energies, as shown below. To illustrate the effect, we only consider orbital interactions between the middle state with the upper and lower ones, with corresponding interaction energies labelled as V_{12} and V_{23} (real and negative numbers). With such, the Hamiltonian that determines the electronic energies with the interactions is given by

$$\begin{bmatrix} E_1 & V_{12} & 0 \\ V_{12} & E_2 & V_{23} \\ 0 & V_{23} & E_3 \end{bmatrix} \quad (\text{R2})$$

To give a quantitative evaluation, we put numbers into equation (R2). For this, we have chosen $E_1 = -3.5$ eV, $E_2 = -5$ eV, $E_3 = -7$ eV, $V_{12} = -1$ eV, and $V_{23} = -1.5$ eV. These energy levels are chosen based on Harrison’s tabulated atomic orbital energies as well as orbital interactions [9], for ZrNiSn (1: Zr, 2: Ni, 3: Sn). We have labeled the energy values on the corresponding orbital lines (Fig. R3a).

Now we take the middle state as an example, which represents the case of the valence band edge state in NbFeSb (see Fig. 2f; though there are more states in practical case, the valence band edge state is the middle one of all the resulting levels). Its interaction with upper level (state E_1) will push itself downwards in energy, endowing it with certain bonding character (Fig. R3b; to illustrate this, imagine only V_{12} is turned on). The energy values after interactions are again indicated above the corresponding orbital lines. On the other hand, the interaction with the lower level will give it an anti-bonding character (Fig. R3c). Now if we turn on both interactions, we will see that as a result of both bonding and anti-bonding interactions, we expect the energy of the middle level to reside in between those values shown in Fig. R3b-c, thereby being closer to the initial orbital energy E_2 (compare the middle level energy shift from -5 eV in (d) with those in (b) and (c)). That is to say, the bonding and anti-bonding interactions cancel each other, in the sense that the middle level energy is barely affected by the existence of the interactions. Such weak dependence on

orbital interaction is favorable for low deformation potential, because the main source of deformation potential comes from the changes of orbital interaction energies (V_{ij}) when atoms are farther apart. If the energy level already does not depend on V_{ij} much, the expansion of lattice then won't have a big effect, translating to low deformation potentials.

Figure R3. (a) Schematic of the starting energy levels and their orbital interactions; (b) Orbital diagram if only interaction between state 1 and state 2 is turned on; (c) Orbital diagram if only interaction between state 2 and state 3 is turned on; (d) Orbital diagram if both interactions between 1 and 2, as well as 2 and 3, are turned on.

To calculate deformation potential, we expand the lattice uniformly. According to Harrison, d-d orbital interaction (V_{23}) varies with atomic distance approximately as $1/d^5$, while d-p orbital interaction (V_{12}) varies as $1/d^{3.5}$. With this, one can calculate the energies for two cell sizes and subtract to obtain deformation potentials.

To quantify the bonding character, we can look at how much it deviates from its initial orbital energy. In the following, we plot the deformation potentials with respect to the energy deviation from its corresponding initial orbital level, for all three states.

Figure R4. Deformation potentials compared with energy difference in a three-level model.

A significant positive (negative) energy shift would mean a strong anti-bonding (bonding) character, which will usually lead to large deformation potentials. This is seen for the upper and lower levels. For the middle level, because its interactions with upper and lower levels relatively cancel, its energy shift is close to zero, thereby leading to a small deformation potential. This quantitative tight-binding model illustrates the idea of how the cancellation of bonding / anti-bonding interactions can give rise to small deformation potentials.

Furthermore, we note that the low deformation potential is not a consequence of d orbitals. As can be seen in Fig. R4 above, the upper level in our model is also a d orbital, but nevertheless has large deformation potential value. The small deformation potential comes from the cancellation, or in other words, the weak dependence on the orbital interactions. Moreover, we mention Mg_2Sn as a specific example where s orbitals can lead to a non-bonding state. It has been known that the conduction band edge state (at X point, with a small group of D_{4h}) in Mg_2Sn is mostly dominated by Mg $3s$ orbital [10]. Our symmetry analysis has shown that s orbital on Mg atoms can form a state that corresponds to the representation of B_{2g} in the D_{4h} point group, which is also a non-bonding orbital (does not share representation with any other state). This is therefore another direct evidence that non-bonding orbitals are not limited to d orbitals. However, as a result of our current numerical limitations (EPW now can only deal with norm-conserving pseudopotentials), we predict this material to be a metal, which has prevented us from evaluating its transport properties. Therefore, we do not present its transport calculations in our work.

We hope our discussion above has explained why when bonding and anti-bonding interactions compete (or, cancel) the resulting level would have a small deformation potential.

Revision: We have added more discussions in the **crystal orbital analysis** section in the manuscript. In particular, we mentioned how the competition of bonding and anti-bonding interactions can lead to a nearly non-bonding level with small deformation potential, as shown above. We have also given the example of Mg_2Sn to specifically show that non-bonding orbitals need not be limited to d orbitals. A new section is also added into the supplementary information (Supplementary Note 3) to elaborate on this three-level tight-binding model.

Comment 3: Also, I disagree with the molecular orbital diagrams shown in Fig. 2. Why don't $d_{x^2-y^2}$ orbitals from atom A and atom B interact? There is no symmetry constraint preventing them from interacting and d -orbitals of Nb are not that confined? The possibility of this interaction conflicts with the explanation offered by the authors.

Response: We apologize that our drawings do not make our definition of orbitals clear. As mentioned in our Methods section (“Symmetry analysis”), the orbitals drawn on the two sides in Fig. 2e and Fig. 2f are not located on a single atom, but instead are a superposition of atomic orbitals on atoms with the same type, with a phase factor that satisfies the Bloch wavefunction form (equation (3) in Methods). For example, the $d_{x^2-y^2}$ orbital on atom A, is interpreted as putting $d_{x^2-y^2}$ orbital on every atom A, combining the phase factors, and then sum up together. This grouped atomic orbital is commonly used in tight-binding (or, linear combination of atomic orbital, LCAO) methods, and is usually called “*crystal orbital*”.

A single $d_{x^2-y^2}$ *atomic* orbital on A can indeed interact with another $d_{x^2-y^2}$ *atomic* orbital on B. However, due to the tetrahedral arrangement of B atoms around A, if we consider several B atoms and sum them up, those interactions eventually cancel. In other words, the $d_{x^2-y^2}$ *crystal* orbital on A would not interact with the $d_{x^2-y^2}$ *crystal* orbital on B. A brief explanation is, when one B atom interacts with A and contributes to some positive interaction energy, another B atom on another tetrahedral side near A would have flipped lobes of the $d_{x^2-y^2}$ orbital, thereby contributing to a negative energy which then cancel. A rigorous and more general argument comes from the symmetry analysis, and this is exactly where the crystal structure comes in to play a role in determining which interaction has to vanish. This is true regardless of the spatial range of the orbitals.

We also note, the symmetry labels shown in Fig. 2a and Fig. 2c are results from DFT calculations. Their agreement with our theoretical diagram shown in Fig. 2e and Fig. 2f is another evidence that our orbital diagram is correct.

Revision: To clarify our analysis, we have expanded the discussion in the manuscript (second paragraph in **crystal orbital analysis** section). We clarified that the starting orbitals we draw in Fig. 2 are *crystal* orbitals, defined according to equation (3). We hope our explanation has addressed the reviewer's question.

Comment 4: Even if we assume that $d_{x^2-y^2}$ orbitals from A and B sites do not interact, the symmetry constraints apply only to the CBM of NbFeSb and it is unclear from the paper whether the VBM formed of d_{xz} and d_{yz} orbitals from the site B, and p_x and p_y from the site C would exhibit similarly weak electron-acoustic phonon interactions or not? Experimental results from Fig. 3 are for the p-type NbFeSb, hence, pertinent to the VBM and not CBM! Analogous question applies to the CBM of ZrNiSn.

Response: The fact that the band edge state has weak electron-acoustic phonon interactions is shown by those calculated deformation potentials as given in Fig. 1 (and also Fig. R2 above). We believe the reviewer's question is why we think the VBM state of NbFeSb (and CBM of ZrNiSn) is a non-bonding state, despite the fact that from the symmetry analysis it seems that this band edge state originates from interactions with many other levels. Another question is how this can lead to low deformation potential.

We want to first clarify that our argument results from a combination of symmetry analysis that leads to the orbital diagram and the projected density-of-states (pDOS). In the case of VBM of NbFeSb, although strictly speaking the band edge state is not non-bonding, the symmetry has prevented its interactions with many other levels with representation A_1 . The result is that we have a band edge state which is a nearly non-bonding level (Fig. 2f). The reason for this is mostly clearly given by the pDOS, which shows that the valence band edge of NbFeSb predominantly comes from the d orbital of atom B. If it had interacted strongly with site C (with lower energy) or site A (with higher energy), the band edge state would have had acquired large projection on C and A's orbitals, which is not the case. We thus believe the resulting state at the VBM of NbFeSb is close to a non-bonding state.

In the case of ZrNiSn, we note that the CBM state corresponds to the upper level formed via orbital interactions within a three-level subset characterized by the representation B_2 (Fig. 2e). In our original drawing, however, this level resides above the non-bonding B_1 state, and should not be the band edge state. Meanwhile, it has a large anti-bonding character, and will also have large deformation potential as shown in Fig. R4 above. To understand its small deformation potential, we note that its large anti-bonding character has to be cancelled by some bonding interactions. This is to happen only when there are additional interactions between B_2 and atomic orbitals at even higher energies (that are not included in our diagram), which push B_2 level lower in energy. The fact that the actual band structure shows B_2 energy level to be lower than B_1 , is a direct evidence for this argument. To illustrate this point, in the three-level system we introduce another energy level (E_0), and for simplicity let it only interact with the initial upper level E_1 state (the corresponding interaction denoted as V_{01}). In this case, the system becomes a four-level system, with Hamiltonian given by

$$\begin{bmatrix} E_0 & V_{01} & 0 & 0 \\ V_{01} & E_1 & V_{12} & 0 \\ 0 & V_{12} & E_2 & V_{23} \\ 0 & 0 & V_{23} & E_3 \end{bmatrix} \quad (\text{R3})$$

We define $E_0 = -1$ eV, and $V_{01} = -2$ eV. Solving this system with and without the V_{01} , we obtain an evolution of the energy levels as a result of this additional interaction, as shown below.

Figure R5. (a) Schematic of the starting energy levels in a three-level system with an additional level; (b) Orbital diagram if interaction between state 0 and state 1 is turned off; (c) Orbital diagram if interaction between state 0 and state 1 is turned on.

Without this additional interaction (Fig. R5b), the upper level B_2 state has energy -2.87 eV and is above its initial orbital energy (-3.5 eV) and thus B_1 state. When the interaction with a higher level is turned on, we see that its energy is pushed lower, below the initial orbital energy (-3.5 eV). This lowering of energy is mainly a consequence of this state acquiring more bonding character due to

its interactions with higher energy levels. The result of this is a cancellation of bonding and its initial anti-bonding character, as clearly seen by its small energy shift (-3.62 eV) from the E_I orbital energy. With this picture in mind, the observation that B_2 state lies below B_I state then implies that now B_2 state should have small deformation potential.

Revision: We have expanded our discussions on the statement that the band edge states for the VBM of NbFeSb and CBM of ZrNiSn are (nearly) non-bonding orbitals (third and fourth paragraph in **crystal orbital analysis** section). We particularly referred to the three-level tight-binding model section in the supplementary information, to give a more quantitative evaluation of how the orbital interactions can lead to low deformation potentials in these cases. We hope our explanations have clarified the reasoning behind the connection between non-bonding orbital and low deformation potentials.

Comment 5: I have hard time imagining that there are no localized spins on Fe or Ni atoms in these systems and from what I can tell non-spin polarized calculations have been performed. Including spin degrees of freedom can significantly affect the electronic band structures and phonon dispersions and could influence the results. I cannot see in the paper what would be the effect of including spins?

Response: We have assumed non-spin polarized configuration in our transport calculations. We have checked that, by initializing the atomic magnetization to be non-zero values, the electronic structures eventually converge to a configuration with zero magnetization, which indicates that the ground state of the materials we studied are non-magnetic. A comparison between the total energies calculated with and without the initialization of magnetization is included, and the results are the same except for the last digit.

Table R1. Comparison of total energies of electronic structure calculations with and without initialization of magnetizations for NbFeSb and ZrNiSn.

Total energy (Ryd)	With initial magnetization	Without magnetization
NbFeSb	-825.23345827	-825.23345829
ZrNiSn	-936.46269776	-936.46269776

Revision: We have added a section in Methods (**Electronic structure calculation**) to discuss our use of non-spin polarized configuration, and have added the above table to the supplementary information.

Comment 6: Also, I am a big proponent of the reproducibility of results and am very much against putting all the technical details in the supplementary materials. The paper needs to be self-contained and the minimal info that allows reproducing results needs to be provided in the main text. Presently, that is not the case, calculations are referred to as first-principles without any further information on what that actually means. And as important, by putting all the info about

the tools in the supplementary section the authors are not giving proper credit to the developers whose tools are being used to obtain the results. This is simply wrong.

Response and Revision: We thank the reviewer for this comment. We have no intention of ignoring the developers' huge efforts and apologize that our arrangement of manuscript and SI did not reflect our appreciation. We have expanded our Methods section so that we can include as many relevant references as possible directly in the manuscript. Some details (mesh, lattice constants, etc.) are kept in SI only as a result of space limitation.

Reviewer #4:

General comment: In the past years, most efforts on half-Heusler TE materials were focused on suppressing the phonon transport, relatively less attention had been paid for the intrinsic reasons leading to their high power factors. Beyond the high band degeneracy that benefits the Seebeck coefficient, low deformation potentials in typical half-Heusler TE materials, such as (Zr,Hf)NiSn, (V,Nb)FeSb, have also been experimentally justified contributing to their relatively high carrier mobilities. However, the intrinsic origin of the low deformation potential is still elusive. The manuscript by Zhou et al. revealed that the origin of low deformation potential in half-Heuslers lies in the weakened electron-acoustic phonon couplings, which further emerge from crystal symmetry-protected non-bonding orbitals. This is a new insight that might be helpful to design new TE materials with enhanced power factor. There are some comments that should be considered by the authors:

Response: We thank the reviewer for finding our work an insightful contribution to the understanding of high power factors, and also for his/her detailed comments on the symmetry argument and transport property simulations. The reviewer's main question is whether our proposed picture of symmetry protected non-bonding orbital is general, or is only limited to *d* orbital dominated system. In response, we state that this should be a general principle and not limited to any particular orbital type. This can be seen by the fact that the symmetry analysis does not make any constraint on the orbital type, as long as their combination with certain crystal structure leads to a non-bonding level. We give one example of Mg₂Sn to specifically show that *s* orbital can also become a non-bonding orbital.

The reviewer's other comments are on the transport properties at higher carrier concentrations and temperatures, which are relevant for thermoelectric materials. Below we provide simulation results at these conditions. Our discussion has been using low carrier density and room temperature results, but the conclusion thus drawn is not limited to these conditions. By presenting data in a larger range, we hope these could stimulate future work into fabricating and testing some of the less-explored compounds to uncover their potential as good thermoelectric materials.

Comment 1: Generally, the conduction band CB and valence band VB of half-Heuslers are dominated by *d* orbitals, which is more localized than *s* or *p* orbitals, and thus more difficult to "deform". My question is whether the non-bonding orbital can only be found in *d* orbital dominated system or can it also exist in the *s* or *p* orbitals dominated system. As the CB and VB of most good thermoelectrics are dominated by *s* and *p* orbitals. More discussion on this point would improve the quality of the manuscript.

Response: We thank the reviewer for this comment. In response, we state that non-bonding orbitals do not have to be limited to *d* orbitals. As long as the symmetry forbids the interaction between a given orbital with others, this orbital should become a non-bonding state. To illustrate that *s* or *p* orbital can also become a non-bonding state, we mention Mg₂Sn as another example. It has been known that the conduction band edge state (at X point, with a small group of D_{4h}) in Mg₂Sn is mostly dominated by Mg 3*s* orbital [10]. Our symmetry analysis has shown that *s* orbital on Mg atoms can form a state that corresponds to the representation of B_{2g} in the D_{4h} point group, which is also a non-bonding orbital (does not share representation with any other state). However, our

band structure calculation of Mg_2Sn predicts it to be a metal, which has been similarly reported by past work [11], and is a result of the inherent limitation of DFT's treatment on electron correlations. Because current implementation of Wannier interpolation can only use norm-conserving pseudopotentials which inevitably leads to a negative band gap for Mg_2Sn , we do not present its transport calculations in this work.

Revision: To clarify that non-bonding orbital is not limited to d orbital-dominated system, we have added discussions on the general condition of the symmetry analysis, as well as the example of Mg_2Sn (second and last paragraph in **crystal orbital analysis** section).

Comment 2: In Figure 3a, the electron scattering rate of ZrNiSn was calculated with a carrier concentration of 10^{18} cm^{-3} . However, the general optimal carrier concentration of Half-Heusler TE materials is in the range of $10^{20} - 10^{21} \text{ cm}^{-3}$. Besides, the optical phonon usually dominates the carrier transport at low carrier concentration, while at higher carrier concentration, the acoustic phonon contributes more (Ref. 25). Adding the electron scattering rate at higher carrier concentration ($10^{20} - 10^{21} \text{ cm}^{-3}$) is highly suggested to identify the dominated carrier transport mechanism in actual half-Heusler TE system.

Response: We have checked that the acoustic phonon scatterings consistently lie lower than optical phonon scatterings in the range of $10^{18} - 10^{21} \text{ cm}^{-3}$. In Fig. R6 we show one example of electron scattering rates as in Fig. 3a but for a carrier concentration of 10^{21} cm^{-3} .

Figure R6. Electron scattering rates decomposed into contributions from different branches for ZrNiSn at room temperature with a carrier concentration of 10^{21} cm^{-3} .

The difference between our results and Ref. 25 is due to the fact that we have specifically excluded polar optical scatterings when comparing different phonon branches. Polar optical phonon scatterings are important for low carrier concentrations, but are often not significant at higher

carrier concentrations because of the carrier screening effect. This is what has been shown by Ref. 25 (there polar scattering is included and therefore at low carrier density optical phonon scattering dominates). By excluding polar scattering, we focus on the deformation potential scattering on electrons, which is most relevant for thermoelectric materials. Therefore, the scattering rates at only 10^{18} cm^{-3} already reflects the order-of-magnitude comparison between acoustic and optical phonon scatterings.

Revision: We have added the scattering rate plot at 10^{21} cm^{-3} into the supplementary information.

Comment 3: In Figure 3c, the authors calculated the optimal power factor at room temperature. While half-Heuslers typically show good TE performance at higher temperature $> 800 \text{ K}$, it is suggested to add the result at higher temperature.

Response: Our work has been focused on the mechanism behind large power factors, and the revealed picture is not limited to the temperature we have chosen. Nonetheless, we agree with the reviewer that presenting higher temperature data will be more helpful especially for experimental groups. One challenge that prevents us from doing such calculations for all materials is that first principles electron transport calculation is very time-consuming, especially at high temperatures. At room temperature, we made use of the fact that only a fraction of electrons residing in Brillouin zone contribute significantly to transport, thereby making our computation feasible. At higher temperatures, many carriers are thermally excited and have to be taken into account in the calculations, leading to much more computational time. We have added high temperature results (1000K) in the revised manuscript as suggested by the reviewer, but have only performed this for a select class of materials due to this computational limit. We realize that some materials in our list have been extensively studied, while some others received much less attention. Therefore, we decide to specifically study the high temperature properties of those less-understood materials, in the hope that this could stimulate more experimental work into fabricating and testing their thermoelectric performance. These include, n-type NbFeSb, p-type ScNiBi, n- and p-type ScNiSb, as well as p-type ZrNiPb.

The optimal power factors for these compounds at 1000K are summarized in the table below. We note that at higher temperatures, the power factors become generally smaller as a result of stronger phonon scatterings. However, n-type NbFeSb, p-type ScNiBi and p-type ScNiSb still exhibit significant power factors.

Table R2. Predicted optimal power factors at 1000K as well as the corresponding carrier concentrations for select half-Heusler compounds.

	n-NbFeSb	p-ScNiBi	n-ScNiSb	p-ScNiSb	p-ZrNiPb
Optimal carrier density (cm^{-3})	$\sim 2e20$	$\sim 4e20$	$\sim 6e20$	$\sim 4e20$	$\sim 6e20$
Optimal power factor ($\mu\text{W}/\text{cm}\cdot\text{K}^2$)	70	70	55	71	42

The electron and phonon mean free path accumulation plot (similar to Fig. 4b-c) at 1000K is also shown here. In general, they become smaller in terms of the mean free path range compared to

room temperature situation. An implication for this is that to efficiently scatter phonons at high temperatures, grain sizes in nanocomposites have to be made even smaller.

Figure R7. (a) The accumulated electrical conductivity with respect to the electron mean free path, and (b) the accumulated thermal conductivity with respect to the phonon mean free path, for select half-Heusler compounds at 1000K.

Revision: We have added discussions regarding high temperature results into the manuscript, and have added Table R2 and Fig. R7 into the supplementary information.

Comment 4: The unit of power factor in the final part of the main text is wrongly written.

Response and Revision: We thank the reviewer for pointing this out. We have corrected the units in our revised manuscript.

Reference

- [1] P. E. Blöchl, “Projector augmented-wave method,” *Phys. Rev. B*, vol. 50, no. 24, pp. 17953–17979, Dec. 1994.
- [2] J. Sjakste, N. Vast, M. Calandra, and F. Mauri, “Wannier interpolation of the electron-phonon matrix elements in polar semiconductors: Polar-optical coupling in GaAs,” *Phys. Rev. B*, vol. 92, no. 5, p. 054307, Aug. 2015.
- [3] M. Lundstrom, *Fundamentals of Carrier Transport*. Cambridge University Press, 2009.
- [4] R. S. Mulliken, “Electronic Population Analysis on LCAO–MO Molecular Wave Functions. I,” *J. Chem. Phys.*, vol. 23, no. 10, pp. 1833–1840, Oct. 1955.
- [5] R. S. Mulliken, “Electronic Population Analysis on LCAO–MO Molecular Wave Functions. II. Overlap Populations, Bond Orders, and Covalent Bond Energies,” *J. Chem. Phys.*, vol. 23, no. 10, pp. 1841–1846, Oct. 1955.
- [6] T. Hughbanks and R. Hoffmann, “Chains of trans-edge-sharing molybdenum octahedra: metal-metal bonding in extended systems,” *J. Am. Chem. Soc.*, vol. 105, no. 11, pp. 3528–3537, Jun. 1983.
- [7] R. Hoffmann, *Solids and surfaces: a chemist’s view of bonding in extended structures*. VCH Publishers, 1988.
- [8] R. Dronskowski and P. E. Bloechl, “Crystal orbital Hamilton populations (COHP): energy-resolved visualization of chemical bonding in solids based on density-functional calculations,” *J. Phys. Chem.*, vol. 97, no. 33, pp. 8617–8624, Aug. 1993.
- [9] W. A. Harrison, *Electronic structure and the properties of solids: the physics of the chemical bond*. Dover Publications, 1989.
- [10] J. J. Pulikkotil *et al.*, “Doping and temperature dependence of thermoelectric properties in $\text{Mg}_2(\text{Si},\text{Sn})$ ” *Phys. Rev. B*, vol. 86, no. 15, p. 155204, Oct. 2012.
- [11] K. Kutorasinski, B. Wiendlocha, J. Tobola, and S. Kaprzyk, “Importance of relativistic effects in electronic structure and thermopower calculations for Mg_2Si , Mg_2Ge , and Mg_2Sn ” *Phys. Rev. B*, vol. 89, no. 11, p. 115205, Mar. 2014.

Reviewers' Comments:

Reviewer #1:

Remarks to the Author:

Since the authors have answered all my comments. I recommend this paper be published in the present form.

Reviewer #3:

Remarks to the Author:

The authors responded appropriately to all of my comments. I am recommending this manuscript for publication.

Reviewer #4:

Remarks to the Author:

With the changes made to the paper I am fully satisfied.

We thank all the reviewers for their positive and constructive comments on the manuscript. In this revised version we have mainly changed the manuscript and supplementary information to meet the format requirements. All the changes are tracked. The revised manuscript and supplementary information are uploaded as separate documents.

Reviewer #1:

Since the authors have answered all my comments. I recommend this paper be published in the present form.

Reviewer #3:

The authors responded appropriately to all of my comments. I am recommending this manuscript for publication.

Reviewer #4:

With the changes made to the paper I am fully satisfied.

Response to all: We thank all reviewers for their positive and constructive comments.